# Very Recent Advances in Vinylogous Mukaiyama Aldol Reactions and Their Applications to Synthesis

**DOI:** 10.3390/molecules24173040

**Published:** 2019-08-22

**Authors:** Martin Cordes, Markus Kalesse

**Affiliations:** Institute of Organic Chemistry, Gottfried Wilhelm Leibniz University of Hannover, Schneiderberg 1b, 30167 Hannover, Germany

**Keywords:** aldol reactions, Mukaiyama, natural product synthesis, stereoselectivity, vinylogy

## Abstract

It is a challenging objective in synthetic organic chemistry to create efficient access to biologically active compounds. In particular, one structural element which is frequently incorporated into the framework of complex natural products is a β-hydroxy ketone. In this context, the aldol reaction is the most important transformation to generate this structural element as it not only creates new C–C bonds but also establishes stereogenic centers. In recent years, a large variety of highly selective methodologies of aldol and aldol-type reactions have been put forward. In this regard, the vinylogous Mukaiyama aldol reaction (VMAR) became a pivotal transformation as it allows the synthesis of larger fragments while incorporating 1,5-relationships and generating two new stereocenters and one double bond simultaneously. This review summarizes and updates methodology-oriented and target-oriented research focused on the various aspects of the vinylogous Mukaiyama aldol (VMA) reaction. This manuscript comprehensively condenses the last four years of research, covering the period 2016–2019.

## 1. Introduction

Aldol reactions are among the most prominent and most frequently applied transformations in synthetic organic chemistry because they assemble the polyketide backbone of important biologically active compounds such as antibiotics and antitumor compounds. These reactions are valuable, not only because they generate new carbon-carbon bonds, but also because they create new stereogenic centers. The most frequently applied methods for aldol reactions often parallel the processes seen in the biosynthesis of polyketide natural products. In the biosynthesis, acetate or propionate units are added; subsequently, a series of further transformations (reductions, eliminations, and hydrogenations) are performed by large polyketide synthases to provide the substrate for the next aldol reaction. The laboratory synthesis mostly follows this modular approach by adding acetate and propionate fragments followed by reduction and oxidation steps, often coupled with extensive protecting group shuffling and additional chain-extension transformations, such as Horner–Wadsworth–Emmons (HWE) reactions (Scheme 1). Even though a wide variety of polyketide structures can be accessed with established transformations, the use of vinylogous aldol reactions reduces the number of steps needed to access these structures. Therefore a substantial number of research groups have focused on these reactions to develop more efficient methods for the construction of larger polyketide segments in one step [1,2,3,4,5]. 

According to Fuson’s principle of vinylogy, an additional adjacent double bond extends the nucleophilic character of silyl enol ethers [6]. Thus, the vinylogous extension of the Mukaiyama aldol reaction allows the synthesis of larger fragments while incorporating 1,5-relationships and generating two new stereocenters and one double bond simultaneously. The vinylogous Mukaiyama aldol (VMA) reaction is of great interest because it provides rapid access to larger carbon frameworks containing a double bond that is available for a wide variety of subsequent transformations (dihydroxylation, epoxidation, cuprate addition, etc.) [5,7,8,9,10,11,12]. The general outcome of the vinylogous Mukaiyama aldol (VMA) reaction is depicted in Scheme 2.

Not surprisingly, Prof. T. Mukaiyama, who first introduced the enoxysilane aldolization chemistry and developed it into a powerful transformation tool, was among the pioneers of the vinylogous version of his eponymous reaction (Scheme 3) [13,14,15].

For the reasons described above, the vinylogous Mukaiyama aldol reaction has become a strategically important and reliable transformation that is increasingly employed in the asymmetric synthesis of complex molecules, particularly polyketides. 

For initiation of the VMA process, two modes of activation have been employed: Activation of the electrophile or activation of the nucleophile (see below). In principle, three general methods are applied to dictate the stereochemical outcome of this transformation: (1) Substrate control, wherein stereoinduction arises from existing stereocenters, (2) catalyst control, wherein an external agent functions as the stereo-controlling element, and (3) a combination of both methods. Because reagent- and substrate-controlled transformations remain dominant in polyketide synthesis, the invention of a general catalyst system to promote catalytic, enantioselective VMA reactions has become a preeminent, overarching goal. 

A brief chronological synopsis of the catalytic, enantioselective VMA reaction, making no claim of being complete, shows that the first examples were reported between 1992–1995 using boron- and titanium(IV)-derived catalysts [16,17,18]. Following this, bisoxazoline–copper(II) complexes and a chiral bisphosphoramide catalyst in conjunction with SiCl_4_ were also shown to be effective in mediating vinylogous Mukaiyama aldol reactions [19,20,21,22,23]. The catalyst systems mentioned thus far represent enantioselective methods for simple aldol reactions extended to vinylogous Mukaiyama aldol reactions; the catalytically active species in these VMA reactions is a chiral Lewis acid, which mediates the reaction by electrophilic aldehyde activation. The first catalyst system that was specifically designed for use in the catalytic, enantioselective, and vinylogous Mukaiyama aldol reaction was a copper(II) fluoride/Tol-BINAP complex in which the catalytically active species was a bisphosphinyl copper(I) fluoride complex [24,25,26].

Since the first pioneering reports of the Mukaiyama aldol reaction itself [27] and the catalytic, enantioselective version of the vinylogous Mukaiyama aldol reaction [16,17], the enantioselective VMA process has established itself as the gold standard for site-selective construction of densely adorned δ-hydroxylated a,β-unsaturated carbonyl compounds and related polyketide networks [1,2,3,5,7,8,9,28].

In contrast to the Mukaiyama aldol reaction, its vinylogous extension, the so-called VMA reaction, involves a regioselectivity issue, namely α-versus γ-addition. In general, metal dienolates favor α-alkylation and silyl dienol ethers favor γ-alkylation products (Scheme 4). Aluminum tris(2,6-diphenylphenoxide) (ATPH) mediated vinylogous aldol reactions are exceptions to this rule, because the α position is deeply buried in the pocket of the catalyst and therefore is no longer accessible to electrophiles [29]. The site selectivity associated with metal dienolates can be overcome by using their silyl derivatives, whose generation can be controlled by the careful choice of catalysts (promoters) and additives. In this context, silyloxy dienes have emerged as superb surrogates for metal dienolates in the vinylogous aldol reaction.

The different site selectivities of the metal dienolates and the silyl dienol ethers can be explained by frontier molecular orbital electron density calculations [30].

For example, the HOMO coefficient and the electrophilic susceptibility value in (1-methoxy-1,3-butadienyloxy)trimethylsilane at C4 is significantly higher than at C2; a kinetic preference for the γ-addition product is therefore predicted. On the other hand, the lithium dienolate of methyl crotonate displays a larger HOMO coefficient and electrophilic susceptibility value at C2 than at C4, rationalizing the observed α-addition product (Scheme 5) [3]. These electronic effects can be altered by steric interactions and/or chelation.

Under typical conditions, most dienoxysilanes are unreactive toward aldehydes without activation [31,32,33,34,35,36]. Thus, catalysis of the VMA reaction is crucial and can proceed by two fundamentally different mechanisms: (a) Aldehyde activation or (b) dienolate activation (Scheme 6).

In the most common mode of aldehyde activation, a Lewis acid or Brønsted acid binds to the oxygen of the carbonyl group (see simple activation, Scheme 7). Because a highly stereoselective outcome requires a high level of transition–structure organization, the binding mode plays an important role in this process. In other words, to induce a large ΔΔG for competing transition structures, different catalysts use different binding motifs. Conformational-biasing interactions, such as CH hydrogen-bonding, π-stacking, or chelation, incorporated individually or in concert with the Lewis acid/aldehyde complex, can contribute to high facial selectivity (Scheme 7).

The following section summarizes and updates methodology-oriented and target-oriented research focused on the various aspects of the VMA reaction. The manuscript comprehensively condenses the last four years of research, covering the period 2016–2019. The discussion concentrates on enantioselective and/or diastereoselective variants that employ chiral ligand control. Purely substrate-controlled reactions leading to chiral adducts are also included. Reactions resulting in achiral products will not be discussed.

## 2. Enantioselective VMA Reactions

### 2.1. Formal Synthesis of a CB2 Agonist Drug Candidate

The isatin core is an important heterocyclic motif present in diverse natural and non-natural products with biological and pharmaceutical activities. Tertiary alcohol **2** is the key intermediate in the synthesis of a potential drug candidate (CB2 agonist) for reducing neuropathic and bone pain.

In this context, an asymmetric vinylogous Mukaiyama aldol reaction (VMAR) protocol enables access to a CB2 agonist building block. Thus, the organocatalyzed VMA reaction of (*E*)-(buta-1,3-dien-1-yloxy)trimethylsilane and 1-(cyclopropylmethyl)-5-methylindoline-2,3-dione using cinchona based thiourea catalyst **1** led to CB2 agonist precursor **2** in good yield and excellent enantioselectivity (94% ee) (Scheme 8) [37]. It is noteworthy that the role of water was crucial for this reaction. In the absence of water almost no conversion occurred, whereas the presence of 3 equiv led to the VMAR product in 75% yield. 

### 2.2. Enantioselective Synthesis of 2,3,5-Trisubstituted Tetrahydrofurans

Substituted tetrahydrofurans are common structural motifs found in several natural products and biologically active compounds. Therefore, in recent years, many efforts have been devoted to the development of stereoselective methods to generate multi-substituted tetrahydrofurans. Thus, a chiral titanium−BINOL catalyst generated in situ from Ti(O*i*-Pr)_4_ and (*R*)-BINOL induced the VMAR of dienediolate **3** and various aldehydes **4** in diethyl ether to furnish the intermediate vinylogous aldol product **5**. That, in turn, was treated with BF_3_•OEt_2_ and a second equivalent of aldehydes **6** in ethyl acetate to give rise to tetrahydrofurans **7** with good overall yield and good to moderate enantioselectivity (Scheme 9) [38]. It is noteworthy that this tandem reaction, a VMAR followed by a Lewis acid-mediated Prins-type cyclization with a second aldehyde, gave rise to 2,3,5-substituted tetrahydrofurans by generating three new σ-bonds and three new stereogenic centers in a one-pot process. 

## 3. Diastereoselective VMA Reactions

### 3.1. Oxazaborolidinone Catalysis

#### 3.1.1. Aetheramide A

The first total synthesis of the highly potent anti-HIV natural product aetheramide A was accomplished using the VMAR protocol utilizing oxazaborolidinone catalysis as a key step. The reaction of silyl dienol ether **8** and vinylogous aldehyde **9** in combination with oxazaborolidinone catalyst **10** gave the secondary alcohol **11** in 89% yield as a single diastereomer (Scheme 10) [39]. The observed diastereoselectivity and the absolute configuration can be explained with a transition state depicted in Scheme 10. Herein, the indole moiety shields the *Si*-face of the aldehyde leading to a *Re*-face attack of the nucleophile.

#### 3.1.2. Nannocystin Ax

Nannocystin Ax is a cytotoxic 21-membered depsipeptide which was isolated from the myxobacterial genus *Nannocystis sp*. In the second total synthesis of Nannocystin Ax, a vinylogous Horner−Wadsworth−Emmons reaction (HWE) and a vinylogous Mukaiyama aldol reaction (VMAR) were used as the key steps for the construction of the polyketide fragment. Thus, aldehyde **13**, TES-ketene acetal **12**, and the oxazaborolidinone Lewis acid **10** generated in situ from *N*Ts-L-tryptophan and dichlorophenylborane were combined in a VMAR. Coordination between the aldehyde and the chiral Lewis acid leads to an attack from the less hindered *re* face of the aldehyde, giving preference to the *R*-configured hydroxy group at C5. Alcohol **14** was obtained in good yield but unfortunately with only moderate diastereoselectivity for the desired product (Scheme 11) [40]. 

### 3.2. The Kobayashi Protocol

In 2004, a vinylogous extension of the Evans’ aldol strategy, the so-called Kobayashi protocol, was investigated. The highly diastereoselective VMA reaction using Evans’ auxiliary-based vinylketene silyl *N*,*O*-acetals provided an efficient and hitherto unprecedentedly high degree of remote [1,7- and 1,6,7-] asymmetric induction [41]. 

#### 3.2.1. Nannocystin A

Nannocystin A is a 21-membered cyclodepsipeptide showing remarkable anti-cancer properties. In 2016 two total syntheses of nannocystin A were described featuring a very similar VMAR according to the Kobayashi protocol [41]. The independent and contemporaneous development of VMAR adducts **17a** and **17b** demonstrates the extreme viability and sustainability of this process (Scheme 12) [42,43]. In a parallel effort, nannocystin A was constructed from essentially the same fragment **18**. 

#### 3.2.2. Nannocystin Ax

The first total synthesis of nannocystin Ax features an extension of the Kobayashi protocol, namely a *syn*-selective VMAR using acetals mediated by BF_3_•OEt_2_ (Scheme 13) [44]. This reaction originally was developed in 2013 by Hosokawa et al. [45]. Thus, the aldol reaction of acetal **19** and vinylketene silyl *N*,*O*-acetal *ent*-**15a** under the action of BF_3_•OEt_2_ directly afforded the methyl ether **20** with 14:1 dr in 88% yield. The advantage of this VMAR is the direct formation of a methoxy group, a feature often required in bioactive natural products; this streamlines natural product synthesis. 

#### 3.2.3. Maltepolide C

Structurally, maltepolide C is a 20-membered cytotoxic macrolide connected to a side chain through an *E*-double bond. The macrolactone core of the molecule contains seven stereocenters, one *E*,*E*-diene unit, one highly substituted *Z*-olefin, one substituted *E*-olefin in conjugation with the lactone carbonyl, and a highly substituted THF moiety. From this structural information and previous studies [46], it was recognized that a Kobayashi VMAR could establish the C1–C7 fragment **23** of the bioactive molecule. Accordingly, treatment of *N*,*O*-silyl ketene acetal **21** and iodo acrylate **16** with TiCl_4_ provided the desired hydroxyl imide **22** with the exclusive formation of one diastereomer in 75% yield (Scheme 14) [47].

#### 3.2.4. (+)-Methynolide

(+)-Methynolide is an aglycone of the 12-membered macrolidic antibiotic methymycin and has been a key target molecule over the past four decades. In the highly convergent total synthesis of (+)-methynolide, the C1–C7 fragment was prepared by a Kobayashi VMAR protocol using a vinyl ketene silyl *N*,*O*-acetal and a β-oxyaldehyde (Scheme 15) [48]. It is known that β-oxyaldehydes are challenging substrates in Kobayashi VMARs, probably due to the β-elimination of an oxygen-containing functional group, and/or chelation that results in the deactivation of the Lewis acid [49,50].

For this purpose, exploratory studies have identified β-oxyaldehyde **24c** (Scheme 15, entry 10) as an ideal candidate for this pivotal transformation, resulting in both excellent yield and diastereoselectivity. The desired *anti*-aldol product **25c** could be obtained in 90% yield on a gramscale using 2.0 equivalent of aldehyde **24c** in toluene in the presence of catalytic amounts of H_2_O [51].

#### 3.2.5. Lagunamide A

Lagunamide A not only possesses excellent anti-malarial properties, but it also has unique, highly cytotoxic properties against leukemia cell lines and against colon cancer. The therapeutic potential of lagunamide A coupled with its relative scarceness in nature has garnered significant interest from a number of research laboratories. In a recent approach, two iterative Kobayashi VMARs have been applied in order to selectively install three contiguous stereocenters at C37, C38, and C39 of the southern polyketide (C27–C45) segment of lagunamide A. The first VMAR [50,52] established the anti-aldol motif at C38–C39 and the second VMAR [50] introduced the stereocenter at C37 (Scheme 16). By adopting a known protocol [52], the first VMAR yielded 96% of anti-alcohol **28** in an excellent diastereomeric ratio (>98:2). The second VMAR, employing aldehyde **29** and chiral vinylketene silyl *N*,*O*-acetal *ent*-**15a**, was more challenging, however. Initially, the VMAR conducted in the conventional CH_2_Cl_2_ medium provided a low yield (30%) of alcohol product **30** albeit with good diastereoselectivity (91:9). However, when toluene containing 10 mol% of water was applied as a medium and the reaction was conducted for 72 h the yield increased modestly to 48% with the same diastereoselectivity (91:9) instead (see entry 2 in the table included in Scheme 16). The exact role of the presence of water during the Kobayashi VMAR is unknown, but the rate enhancement effect of H_2_O was demonstrated in previous studies [51].

#### 3.2.6. Amphidinolide N

Amphidinolide N is an extremely potent cytotoxic macrolide that was first isolated from a cultured *Amphidinium sp.* (Y-5) strain in 1994. Due to its challenging structure and extraordinarily potent cytotoxicity, amphidinolide N is a highly attractive target for total synthesis. Consequently, a convergent synthesis of the C1–C13 segment of amphidinolide N was reported using a Kobayashi VMAR as a key step (Scheme 17) [53]. For that, the VMAR of acetaldehyde **32** with the dienol silyl ether **31** delivered δ-hydroxy-α-methyl-α,β-unsaturated imide **33 [54]** in 91% yield as a single diastereomer (dr > 20:1).

#### 3.2.7. Tabtoxinine-β-Lactam (TβL)

Tabtoxinine-β-lactam (TβL) is a potent inhibitor of glutamine synthetase. Therefore, TβL is expected to be a selective pesticide and many synthetic studies on TβL have been reported, recently cumulating in a stereoselective synthesis of TβL by a remote asymmetric induction. The pivotal Kobayashi VMAR process is depicted in Scheme 18 [55]. 

Here, careful tuning of the chiral auxiliary (R_2_), the silyl ether group (R_1_), and the Lewis acid afforded tertiary alcohol **36** in excellent yield and selectivity (see entry 5 in the table included in Scheme 18).

#### 3.2.8. Fidaxomicin

Fidaxomicin, also known as tiacumicin B and lipiarmycin A3, constitutes a macrolide antibiotic used for the treatment of *Clostridium difficile* infections (CDI), which are considered to be responsible for a significant number of hospital-acquired infections, resulting in over 29000 deaths each year in the US. While vancomycin and metronidazole are generally prescribed for this condition as a first-line treatment, the introduction of fidaxomicin in the USA and the EU in 2011 provided a new therapeutic option. Due to increased demand, the commercial macrolide antibiotic fidaxomicin very recently was synthesized in a highly convergent manner using a Kobayashi VMAR as a key step. 

Interestingly, the described VMAR protocol requires a more sophisticated procedural approach. In the first generation synthesis of **38** (entry 1, Scheme 19) [56], a premixed solution of TiCl_4_ and aldehyde **16** at –78 °C was treated with vinylketene silyl-*N*,*O*-acetal **37**, and added in one portion at –78 °C. In the second generation synthesis and after extensive experimentation, it was found that aminal **37** does not need to be added in one portion, instead it can be delivered by syringe pump at –78 °C over a period of 20 min to cleanly obtain the desired product in high yield and excellent diastereoselectivity (entry 2, Scheme 19) [57]. It is noteworthy that for the transformation of **38** to the aglycone of fidaxomicin the stereochemistry at C11 had to be inverted by a Mitsunobu reaction.

#### 3.2.9. PF1163B

Recently, a concise total synthesis of PF1163B was achieved (Scheme 20) [58]. For this, Hosokawa’s extension [59] of the Kobayashi VMAR of ketene silyl *N*,*O*-acetal **15a**, mediated by excess TiCl_4_ (4 equivalent), proceeded with saturated aldehyde **39** to give adduct **40** in moderate yield with moderate stereoselectivity. The Birch reduction of α,β-unsaturated imide **40**, possessing the *R* configured, less hindered isopropyl side chain, gave the reduction product **41** with the *S* configuration at C10 in good stereoselectivity (6:1) by employing 2-isopropylbenzimidazole as a bulky proton source. 

Of note, in the described VMAR protocol, two separate stereogenic centers in PF1163B, C10, and C13 respectively, were constructed with one chiral synthon. This demonstrates a short sequence to prepare medium-size derivatives.

#### 3.2.10. Stoloniferol B

Stoloniferol B, isolated from *Penicillium stoloniferum* QY2-10, is a common core structure of citrinin derivatives. However, the total synthesis of stoloniferol B had never been reported. The first total synthesis of stoloniferol B was presented using a Kobayashi VMAR. The synthesis started with the vinylogous Mukaiyama aldol reaction between **21** and paraldehyde **42** (Scheme 21) [60]. The reaction proceeded in high yield (82%) with excellent diastereoselectivity (>20:1) to give *anti* adduct **43**.

#### 3.2.11. C1–C17 Segment of Bafilomycin N

Because bafilomycin A_1_ is a well-known anti-tumor agent inhibiting v-ATPase at picomolar concentration, its congener bafilomycin N is also expected to be an attractive target for antitumor drugs. Although total synthesis of bafilomycin A_1_ has been achieved, synthetic studies on the new congener bafilomycin N are unreported. Very recently, the first total synthesis of the C1–C17 segment of bafilomycin N was presented using an *anti*- (C3–C11) and a *syn*- (C12–C17) selective Kobayashi VMAR (Scheme 22) [61]. Strikingly, the two fragments **45** and **47** were constructed with the same chiral synthon. Thus, aldehyde **44** was subjected to the vinylogous Mukaiyama aldol reaction with the use of *ent*-**21** to give *anti* adduct **45** (C3–C11). The rate enhancement effect of H_2_O in the VMAR was demonstrated in previous studies [51]. The synthesis of C12–C17 segment **47** started from the *syn*-selective Kobayashi VMAR between vinylketene silyl *N*,*O*-acetal *ent*-**21** and chloral **46**. The reaction proceeded smoothly to afford *syn* adduct **47** in excellent diastereoselectivity (>20:1) [59,62].

#### 3.2.12. C3–C21 Segment of Aflastatin A

Aflastatin A remains an attractive and challenging target molecule because the complexity and large size of this compound require efficient methodologies and adequate strategies to achieve its total synthesis. The very recent synthesis of the C3–C21 segment of aflastatin A involves two Kobayashi VMA reactions as key steps (Scheme 23) [63]. It is noteworthy to mention, the two fragments **49** and *ent*-**43** were constructed with the same chiral synthon. The known VMAR between *N*,*O*-acetal **21** and methacrolein (**48**) has always been a cumbersome process, because under standard conditions (1.0 equivalent of TiCl_4_ in CH_2_Cl_2_, 2.0 equivalent of **48**), the aldol adduct **49** is only formed in low yield (23%) [64], presumably as a result of polymerization of **48**. During an extensive survey of reaction conditions, the authors improved the yield of **49** to 65% with greater than 20:1 diastereoselectivity by using toluene as the solvent and 4.0 equivalent of **48** [64]. In the very recent version of this Kobayashi VMAR a yield of 76% was achieved (Scheme 23) [63]. The second Kobayashi VMAR was also developed earlier [60], and herein repeated as the enantiomeric process with nearly the same yield (80%) and diastereoselectivity (18:1) affording *ent*-**43** [63].

## 4. Conclusions

The efficient synthesis of complex natural products is often a hurdle that has to be overcome before issues of chemical biology can be addressed. On that background, the vinylogous Mukaiyama aldol reaction has become one of the pivotal transformations to gain rapid access to natural products. 

Over the years, a variety of different enantioselective and diastereoselective transformations have been demonstrated that allow the synthesis of all structural motifs found in natural products and polyketides in particular. Besides the improvement of existing methods, one can expect elaboration of subsequent transformations in order to further functionalize the structural motifs which are generated by the vinylogous Mukaiyama aldol reaction.

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
