# Peer review of "Very Recent Advances in Vinylogous Mukaiyama Aldol Reactions and Their Applications to Synthesis"

_molecules, 2019, doi:10.3390/molecules24173040_

Round 1

Reviewer 1 Report

This is a concise and good review on recent examples of the vinylogous Mukaiyama aldol reaction. There is a good introduction and the chosen cases are quite illustrative. Therefore, I recommend acceptance of the manuscript after some minor modifications.

Scheme 11: I think structure 13 is identical to 9.

Scheme 12: One might briefly mention from which seco compound and by which reaction the macrocycle was formed.

Line 287: The sentence "The rate enhancement effect of H2O …" is listed a couple times in the manuscript. I think to mention it once or twice should be enough.

Reference section: Some journal names should be abbreviated (Line 538, 539, 545, 546 etc).

From a scholarly perspective the depiction of one or two transition states would be helpful.

Author Response

Scheme 11: I think structure 13 is identical to 9.

Structure 13 is not identical to 9.

Scheme 12: One might briefly mention from which seco compound and by which reaction the macrocycle was formed.

Yes, one could, but the review describes the VMA reaction and not macrocyclings. We would like to set this aside.

Line 287: The sentence "The rate enhancement effect of H2O …" is listed a couple times in the manuscript. I think to mention it once or twice should be enough.

We agree and will change this.

Reference section: Some journal names should be abbreviated (Line 538, 539, 545, 546 etc).

we corrected that

From a scholarly perspective the depiction of one or two transition states would be helpful.

a reasonable rationale for the steriochemical outcome of the VMAR could only be given for the examples in scheme 10 and 12 (Kobayashi)

Reviewer 2 Report

The review reports the advances of the vinylogous Mukaiyama aldol reactions and their application in stereoselective synthesis with an emphasis on natural product synthesis. The period 2016-2019 was covered. In the introduction, the a and b regioselectivity of the VMAR is discussed but examples have not been presented in the discussion for the reactions exploiting b regioselectivity. The catalyst-controlled VMARs are discussed briefly in the introduction with proper citations but the corresponding paragraph is quite short in the discussion. Only two examples are presented. The authors may improve this paragraph further even at the expense of reducing the examples on the Kobayashi protocol, which is explained by numerous examples.

Similarly, the different activation modes of the carbonyl group is explained in general but the activation modes were not supported by specific examples in the discussion.

The following points needs amendment:

1)      p 5, par 1: „Reactions of achiral products will not be discussed.” Reactions resulting in achiral products….

2)      p 8, par 1: „giving preferentially the R-configured hydroxy group at C7.” C-5 is the chirality center.

3)      The name of the natural product should not be written with capital first letter.

4)      Scheme 13: There is no need to repeat the structure of nannocystin Ax, which is already shown in Scheme 11.

5)      p 10, par 1: acetals instead of actals

6)      Language and style need polishing: e.g.

p 10: „…this makes natural product synthesis speedy.”

p 16: „…requires a more sophisticated procedural management.”

p 16: „…found that aminal 37 not has to be added in one portion..”

7)      p 17: „This is a short sequence to prepare medium-size derivatives.” instead of „This is a short strategy to prepare mediumsize compounds.”

8)      Scheme 21: The absolute configuration of C-3 should be indicated by the orientation of the methyl group, even though this may not help identifying the correlation with compound 43.

Author Response

1)      p 5, par 1: „Reactions of achiral products will not be discussed.” Reactions resulting in achiral products…we have corrected that

2)      p 8, par 1: „giving preferentially the R-configured hydroxy group at C7.” C-5 is the chirality center. we have corrected that

3)      The name of the natural product should not be written with capital first letter.

we have corrected that

4)      Scheme 13: There is no need to repeat the structure of nannocystin Ax, which is already shown in Scheme 11.

you need this structure again for clarification, because other/different carbon atoms are built up

5)      p 10, par 1: acetals instead of actals

we have corrected that

6)      Language and style need polishing: e.g.

p 10: „…this makes natural product synthesis speedy.”

p 16: „…requires a more sophisticated procedural management.”

p 16: „…found that aminal 37 not has to be added in one portion..”

7)      p 17: „This is a short sequence to prepare medium-size derivatives.” instead of „This is a short strategy to prepare mediumsize compounds.”we have corrected that

8)      Scheme 21: The absolute configuration of C-3 should be indicated by the orientation of the methyl group, even though this may not help identifying the correlation with compound 43.

we have corrected that

Reviewer 3 Report

The manuscript seems complete and well-written, and therefore I recommend its publication.

Author Response

ok

Reviewer 4 Report

This is a review manuscript for the vinylogous Mukaiyama aldol reaction. The recent progress and its application using the titled reaction is well summarized. This is a useful information in the research area of synthetic organic chemistry and worth to be published as a review article in Molecules.

Comments are as follows:

Title: “Very” is not necessarily.

Systematic numbering for all the compounds in Schemes and Figures as well as the main text is needed. These aid smooth reading.

The proposed transition state models for the examples in scheme 8, 9, 10, and one of the Kobayashi’s examples is needed.

The key recent reviews must be sited, if these are not appeared in the reference section.

The recent development of vinylogous Mukaiyama aldol reactions S Hosokawa - Tetrahedron letters, 2018 – Elsevier

Development and Application of Asymmetric Organocatalytic Mukaiyama and Vinylogous Mukaiyama‐Type Reactions M Frías, W Cieślik, A Fraile… - … A European Journal, 2018

K. M. Byrd, Beilstein J. Org. Chem. 2015, 11, 530.

 P1

L12 the aldol reaction --- => the aldol reaction has attracted much attention for the generation of this structural motif.

L16 since it allows rapid generation of the vinylogous analog of 1,3-diketone motif.

L31 by large polyketide -- => polyketide ---

L32 The laboratory mostly follows --- => One of the practical approaches involves

L37 Therefore a substantial number of research groups have focused on these reactions in the --- => Therefore a substantial number of research groups have focused on the development of

Author Response

Title: “Very” is not necessarily.

sorry, we can not skip that. publishing 2019 stands for very recent papers

Systematic numbering for all the compounds in Schemes and Figures as well as the main text is needed. These aid smooth reading.

please note, all relevant molecules discussed in the text are numbered

The proposed transition state models for the examples in scheme 8, 9, 10, and one of the Kobayashi’s examples is needed.

a reasonable rationale for the steriochemical outcome of the VMAR could only be given for the examples in scheme 10 and 12 (Kobayashi)

The key recent reviews must be sited, if these are not appeared in the reference section.

the key reviews have been cited

The recent development of vinylogous Mukaiyama aldol reactions S Hosokawa - Tetrahedron letters, 2018 – Elsevier

cited

Development and Application of Asymmetric Organocatalytic Mukaiyama and Vinylogous Mukaiyama‐Type Reactions M Frías, W Cieślik, A Fraile… - … A European Journal, 2018

cited

K. M. Byrd, Beilstein J. Org. Chem. 2015, 11, 530.

Not relevant, Conjugate 1,4  Michael additions, no VMAR processes

 P1

we don't want to change anything, because especially this part of the review was checked by several native speakers (all US)

L12 the aldol reaction --- => the aldol reaction has attracted much attention for the generation of this structural motif.

L16 since it allows rapid generation of the vinylogous analog of 1,3-diketone motif.

L31 by large polyketide -- => polyketide ---

L32 The laboratory mostly follows --- => One of the practical approaches involves

L37 Therefore a substantial number of research groups have focused on these reactions in the --- => Therefore a substantial number of research groups have focused on the development of